# Optimal Analysis of Subset-Selection Based $\ell_p$ Low-Rank Approximation

**Chen Dan**
Carnegie Mellon University
cdan@cs.cmu.edu

**Hong Wang**[*]
Princeton University
Hong.Wang1991@gmail.com

**Hongyang Zhang**[*]
Toyota Technological Institute at Chicago
honyanz@ttic.edu

**Yuchen Zhou**[*]
University of Wisconsin, Madison
yuchenzhou@stat.wisc.edu

**Pradeep Ravikumar**
Carnegie Mellon University
pradeepr@cs.cmu.edu

## Abstract

We study the low rank approximation problem of any given matrix $A$ over $\mathbb{R}^{n \times m}$ and $\mathbb{C}^{n \times m}$ in entry-wise $\ell_p$ loss, that is, finding a rank-$k$ matrix $X$ such that $\|A - X\|_p$ is minimized. Unlike the traditional $\ell_2$ setting, this particular variant is NP-Hard. We show that the algorithm of column subset selection, which was an algorithmic foundation of many existing algorithms, enjoys approximation ratio $(k + 1)^{1/p}$ for $1 \le p \le 2$ and $(k + 1)^{1-1/p}$ for $p \ge 2$. This improves upon the previous $O(k + 1)$ bound for $p \ge 1$ [1]. We complement our analysis with lower bounds; these bounds match our upper bounds up to constant 1 when $p \ge 2$. At the core of our techniques is an application of *Riesz-Thorin interpolation theorem* from harmonic analysis, which might be of independent interest to other algorithmic designs and analysis more broadly.

As a consequence of our analysis, we provide better approximation guarantees for several other algorithms with various time complexity. For example, to make the algorithm of column subset selection computationally efficient, we analyze a polynomial time bi-criteria algorithm which selects $O(k \log m)$ columns. We show that this algorithm has an approximation ratio of $O((k + 1)^{1/p})$ for $1 \le p \le 2$ and $O((k + 1)^{1-1/p})$ for $p \ge 2$. This improves over the best-known bound with an $O(k + 1)$ approximation ratio. Our bi-criteria algorithm also implies an exact-rank method in polynomial time with a slightly larger approximation ratio.

## 1 Introduction

Low rank approximation has wide applications in compressed sensing, numerical linear algebra, machine learning, and many other domains. In compressed sensing, low rank approximation serves as an indispensable building block for data compression. In numerical linear algebra and machine learning, low rank approximation is the foundation of many data processing algorithms, such as PCA. Given a data matrix $A \in \mathbb{F}^{n \times m}$, low rank approximation aims at finding a low-rank matrix

---

[*]Equal Contribution

$X \in \mathbb{F}^{n \times m}$ such that

$$\mathsf{OPT} = \min_{X : \mathrm{rank}(X) \le k} \|X - A\|. \tag{1}$$

Here the field $\mathbb{F}$ can be either $\mathbb{R}$ or $\mathbb{C}$. The focus of this work is on the case when $\|\cdot\|$ is the entry-wise $\ell_p$ norm, and we are interested in an estimate $\widehat{X}$ with a *tight* approximation ratio $\alpha$ so that we have the guarantee: $\|\widehat{X} - A\| \le \alpha \cdot \mathsf{OPT}$.

As noted earlier, such low-rank approximation is a fundamental workhorse of machine learning. The key reason to focus on approximations with respect to general $\ell_p$ norms, in contrast to the typical $\ell_2$ norm, is that these general $\ell_p$ norms are better able to capture a broader range of realistic noise in complex datasets. For example, it is well-known that the $\ell_1$ norm is more robust to the sparse outlier [2–4]. So the $\ell_1$ low-rank approximation problem is a robust version of the classic PCA which uses the $\ell_2$ norm and has received tremendous attentions in machine learning, computer vision and data mining [5], [6], [7]. A related problem $\ell_p$ linear regression has also been studied extensively in the statistics community, and these two problems share similar motivation. In particular, if we assume a statistical model $A_{ij} = A_{ij}^\star + \varepsilon_{ij}$, where $A^\star$ is a low rank matrix and $\varepsilon_{ij}$ are i.i.d. noise, the different values of $p$ correspond to the MLE of different noise distributions, say $p = 1$ for Laplacian noise and $p = 2$ for Gaussian noise.

While it has better empirical and statistical properties, the key bottleneck to solving the problem in (1) is computational, and is known to be NP-hard in general. For example, the $\ell_1$ low-rank approximation is NP-hard to solve exactly even when $k = 1$ [8], and is even hard to approximate with large error under the Exponential Time Hypothesis [9]. [10] proved the NP-hardness of the problem when $p = \infty$. A recent work [11] proves that the problem has no constant factor approximation algorithm running in time $O(2^{k^\delta})$ for a constant $\delta > 0$, assuming the correctness of Small Set Expansion Hypothesis and Exponential Time Hypothesis. The authors also proposed a PTAS (Polynomial Time Approximation Scheme) with $(1 + \varepsilon)$ approximation ratio when $0 < p < 2$. However, the running time is as large as $O(n^{\mathsf{poly}(k/\varepsilon)})$.

Many other efforts have been devoted to designing approximation algorithms in order to alleviate the computational issues of $\ell_p$ low-rank approximation. One promising approach is to apply subgradient descent based methods or alternating minimization [12]. Unfortunately, the loss surface of problem (1) suffers from saddle points even in the simplest $p = 2$ case [13], which might be arbitrarily worse than $\mathsf{OPT}$. Therefore, they may not work well for the low-rank approximation problem as these local searching algorithms may easily get stuck at bad stationary points without any guarantee.

Instead, we consider another line of research—the heuristic algorithm of column subset selection (CSS). Here, the algorithm proceeds by choosing the best $k$ columns of $A$ as an estimation of column space of $X$ and then solving an $\ell_p$ linear regression problem in order to obtain the optimal row space of $X$. See Algorithm 1 for the detailed procedure. Although the vanilla form of the subset selection based algorithm also has an exponential time complexity in terms of the rank $k$, it can be slightly modified to polynomial time bi-criteria algorithms which selects more than $k$ columns [1]. Most importantly, these algorithms are easy to implement and runs fast with nice empirical performance. Thus, subset selection based algorithms might seem to effectively alleviate the computational issues of problem (1). The caveat however is that CSS might seem like a simple heuristic, with potentially a very large worst-case approximation ratio $\alpha$.

In this paper, we show that CSS yields surprisingly reasonable approximation ratios, which we also show to be tight by providing corresponding lower bounds, thus providing a strong theoretical backing for the empirical observations underlying CSS.

Due in part to its importance, there has been a burgeoning set of recent analyses of column subset selection. In the traditional low rank approximation problem with Frobenious norm error (the $p = 2$ case in our setting), [14] showed that CSS achieves $\sqrt{k+1}$ approximation ratio. The authors also showed that the $\sqrt{k+1}$ bound is tight (both upper and lower bounds can be recovered by our analysis). [15–18] improved the running time of CSS with different sampling schemes while preserving similar approximation bounds. The CSS algorithm and its variants are also applied and analyzed under various different settings. For instance, [19] and [20] studied the CUR decomposition with the Frobenius norm. [21] studied the CSS problem under the missing-data case. With $\ell_1$ error, [22] studied CSS for non-negative matrices in $\ell_1$ error. [23] gave tight approximation bounds

for CSS under finite-field binary matrix setting. Furthermore, [9] considered the low rank tensor approximation with the Frobenius norm.

Despite a large amount of work on the subset-selection algorithm and the $\ell_p$ low rank approximation problem, many fundamental questions remain unresolved. Probably one of the most important open questions is: what is the *tight* approximation ratio $\alpha$ for the subset-selection algorithm in the $\ell_p$ low rank approximation problem, up to a constant factor? In [1], the approximation ratio is shown to be upper bounded by $(k+1)$ and lower bounded by $O(k^{1-\frac{2}{p}})$ when $p > 2$. This problem becomes even more challenging when one requires the approximation ratio to be tight up to factor 1, as little was known about a direct tool to achieve this goal in general. In this work, we improve both upper and lower bounds in [1] to optimal when $p > 2$. Note that our bounds are still applicable and improve over [1] when $1 < p < 2$, but there is an $O(k^{\frac{2}{p}-1})$ gap between the upper and lower bounds.

## 1.1 Our Results

The best-known approximation ratio of subset selection based algorithms for $\ell_p$ low-rank approximation is $O(k+1)$ [1]. In this work, we give an improved analysis of this algorithm. In particular, we show that the Column Subset Selection in Algorithm 1 is a $c_{p,k}$-approximation, where

$$c_{p,k} = \begin{cases} (k+1)^{\frac{1}{p}}, & 1 \leq p \leq 2, \\ (k+1)^{1-\frac{1}{p}}, & p \geq 2. \end{cases}$$

This improves over Theorem 4 in [1] which proved that the algorithm is an $O(k+1)$-approximation, for all $p \geq 1$. Below, we state our main theorem formally:

**Theorem 1.1** (Upper bound). *The subset selection algorithm in Algorithm 1 is a $c_{p,k}$-approximation.*

Our proof of Theorem 1.1 is built upon novel techniques of Riesz-Thorin interpolation theorem. In particular, with the proof of special cases for $p = 1, 2, \infty$, we are able to interpolate the approximation ratio of all intermediate $p$'s. Our techniques might be of independent interest to other $\ell_p$ norm or Schatten-$p$ norm related problem more broadly. See Section 1.2 for more discussions.

We also complement our positive result of subset selection algorithm with a negative result. Surprisingly, our upper bound matches our lower bound exactly up to constant 1 for $p \geq 2$. Below, we state our negative results formally:

**Theorem 1.2** (Lower bound). *There exist infinitely many different values of k, such that approximation ratio of any k-subset-selection based algorithm is at least $(k+1)^{1-\frac{1}{p}}$ for $\ell_p$ rank-k approximation.*

Note that our lower bound strictly improves the $(k+1)^{1-\frac{2}{p}}$ bound in [1]. The main idea of the proof can be found in Section 1.2 and we put the whole proof of Theorem 1.2 in Appendix 3.

One drawback of Algorithm 1 is that the running time scales exponentially with the rank $k$. However, it serves as an algorithmic foundation of many existing computationally efficient algorithms. For example, a bi-criteria variant of this algorithm runs in polynomial time, only requiring the rank parameter to be a little over-parameterized. Our new analysis can be applied to this algorithm as well. Below, we state our result informally:

**Theorem 1.3** (Informal statement of Theorem 4.1). *There is a bi-criteria algorithm which runs in $\mathsf{poly}(m, n, k)$ time and selects $O(k \log m)$ columns of A. The algorithm is an $O(c_{p,k})$-approximation algorithm.*

Our next result is a computationally-efficient, *exact-rank* algorithm with slightly larger approximation ratio. Below, we state our result informally:

**Theorem 1.4** (Informal statement of Theorem 4.2). *There is an algorithm which solves problem* (1) *and runs in $\mathsf{poly}(m, n)$ time with an $O(c_{p,k}^3 k \log m)$-approximation ratio, provided that $k = O(\frac{\log n}{\log \log n})$.*

## 1.2 Our Techniques

In this section, we give a detailed discussion about our techniques in the proofs. We start with the analysis of approximation ratio of *column subset selection* algorithm.

---

**Algorithm 1** A $c_{p,k}$ approximation to problem (1) by column subset selection.

---

1: **Input:** Data matrix $A$ and rank parameter $k$.
2: **Output:** $X \in \mathbb{R}^{n \times m}$ such that $\mathsf{rank}(X) \leq k$ and $\|X - A\|_p \leq c_{p,k} \cdot \mathsf{OPT}$.
3: **for** $I \in \{S \subseteq [m]; |S| = k\}$ **do**
4:    $U \leftarrow A_I$.
5:    Run $\ell_p$ linear regression over $V$ that minimizes the loss $\|A - UV\|_p$.
6:    Let $X_I = UV$
7: **end for**
8: **return** $X_I$ which minimizes $\|A - X_I\|_p$ for $I \in \{S \subseteq [m]; |S| = k\}$.

---

**Remark** Throughout this paper, we state the theorems for real matrices. The results can be naturally generalized for complex matrices as well.

**Notations:** We denote by $A \in \mathbb{R}^{n \times m}$ the input matrix, and $A_i$ the $i$-th column of $A$. $A^* = UV$ is the optimal rank-$k$ approximation, where $U \in \mathbb{R}^{n \times k}, V \in \mathbb{R}^{k \times m}$. $\Delta_i = A_i - UV_i$ is the error vector on the $i$-th column, and $\Delta_{li}$ the $l$-th element of vector $\Delta_i$. For any $X \in \mathbb{R}^{n \times k}$, define the error of projecting $A$ onto $X$ by $\mathsf{Err}(X) = \min_{Y \in \mathbb{R}^{k \times m}} \|A - XY\|_p$. Let $J = (j_1, \cdots, j_k) \in [m]^k$ be a subset of $[m]$ with cardinality $k$. We denote $X_J$ as the following column subset in matrix $X$: $X_J = [X_{j_1}, X_{j_2}, \cdots, X_{j_k}]$. Similarly, we denote by $X_{dJ}$ the following column subset in the $d$-th dimension of matrix $X$: $X_{dJ} = [X_{dj_1}, X_{dj_2}, \cdots, X_{dj_k}]$. Denote by $J^*$ the column subset which gives smallest approximation error, i.e., $J^* = \mathrm{argmin}_{J \in [m]^k} \mathsf{Err}(A_J)$.

**Analysis in Previous Work:** In order to show that the column subset selection algorithm gives an $\alpha$-approximation, we need to prove that

$$\mathsf{Err}(A_{J^*}) \leq \alpha \|\Delta\|_p. \tag{2}$$

Directly bounding $\mathsf{Err}(A_{J^*})$ is prohibitive. In [1], the authors proved an upper bound of $(k+1)$ in two steps. First, the authors constructed a specific $S \in [m]^k$, and upper bounded $\mathsf{Err}(A_{J^*})$ by $\mathsf{Err}(A_S)$. Their construction is as follows: $S$ is defined as the minimizer of

$$S = \operatorname*{argmin}_{J \in [m]^k} \frac{|\det(V_J)|}{\prod_{j \in J} \|\Delta_j\|_p}.$$

In the second step, [1] upper bounded $\mathsf{Err}(A_S)$ by considering the approximation error on each column $A_i$, and upper bounded the $\ell_p$ distance from $A_i$ to the subspace spanned by $A_S$ using triangle inequality of $\ell_p$ distance. They showed that the distance is at most $(k+1)$ times of $\|\Delta_i\|$, uniformly for all columns $i \in [m]$. Therefore, the approximation ratio is bounded by $(k+1)$. Our approach is different from the above analysis in both steps.

**Weighted Average:** In the first step, we use a so-called *weighted average* technique, inspired by the approach in [14, 23]. Instead of using the error of one specific column subset as an upper bound, we use a weighted average over all possible column subsets, i.e.,

$$\mathsf{Err}^p(A_{J^*}) \leq \sum_{J \in [m]^k} w_J \mathsf{Err}^p(A_J),$$

where the weight $w_J$'s are carefully chosen for each column subset $J$. This weighted average technique captures more information from all possible column subsets, rather than only from one specific subset, and leads to a tighter bound.

**Riesz-Thorin Interpolation Theorem:** In the second step, unlike [1] which simply used triangle inequality to prove the upper bound, our technique leads to more refined analysis of upper bounds for the approximation error for each subset $J \in [m]^k$. With the technique of weighted average in the first step, proving a technical inequality (Lemma 2.2) concerning the determinants suffices to complete the analysis of approximation ratio. In the proof of this lemma, we introduce several powerful tools from harmonic analysis, the theory of interpolating linear operators. Riesz-Thorin theorem is a classical result in interpolation theory that gives bounds for $L^p$ to $L^q$ operator norm. In general, it is easier to prove estimates within spaces like $L^2$, $L^1$ and $L^\infty$. Interpolation theory enables us to generalize

results in those spaces to some $L^p$ and $L^q$ spaces with an explicit operator norm. By the Riesz-Thorin interpolation theorem, we are able to prove the lemma by just checking the special cases $p = 1, 2, \infty$, and then interpolate the inequality for all the intermediate value of $p$'s.

**Lower Bounds:** We now discuss the techniques in proving the lower bounds. Our proof is a generalization of [14], which shows that for the special case $p = 2$, $\sqrt{k+1}$ is the best possible approximation ratio. Their proof for the lower bound is constructive: they constructed a $(k+1) \times (k+1)$ matrix $A$, such that using any $k$-subset leads to a sub-optimal solution by a factor no less than $(1 - \varepsilon)\sqrt{k+1}$. However, since $\ell_p$ norm is not rotationally-invariant in general, it is tricky to generalize their analysis to other values of $p$'s. To resolve the problem, we use a specialized version of their construction, the perturbed Hadamard matrices (see Section 3 for details), as they have nice symmetricity and are much easier to analyze. We give an example of special case $k = 3$ for better intuition:

$$A = \begin{pmatrix} \varepsilon & \varepsilon & \varepsilon & \varepsilon \\ 1 & 1 & -1 & -1 \\ 1 & -1 & 1 & -1 \\ 1 & -1 & -1 & 1 \end{pmatrix}.$$

Here $\varepsilon$ is a positive constant close to 0. We note that $A$ is very close to a rank-3 matrix: if we replace the first row by four zeros, then it becomes rank-3. Thus, the optimal rank-3 approximation error is at most $(4\varepsilon^p)^{1/p} = 4^{1/p}\varepsilon$. Now we consider the column subset selection algorithm. For example, we use the first three columns $A_1, A_2, A_3$ to approximate the whole matrix — the error only comes from the fourth column. We can show that when $\varepsilon$ is small, the projection of $A_4$ to span $\{A_1, A_2, A_3\}$ is very close to

$$-A_1 - A_2 - A_3 = (-3\varepsilon, -1, -1, 1)^T.$$

Therefore, the column subset selection algorithm achieve about $4\varepsilon$ error on this matrix, which is a $4^{1 - \frac{1}{p}}$ factor from being optimal. The similar construction works for any integer $k = 2^r - 1, r \in \mathbb{Z}^+$, where the lower bound is replaced by $(k+1)^{1 - \frac{1}{p}}$, also matches with our upper bound exactly when $p \geq 2$.

## 2 Analysis of Approximation Ratio

In this section, we will prove Theorem 1.1. Recall that our goal is to bound $\mathsf{Err}(A_{J^*})$. We first introduce two useful lemmas. Lemma 2.1 gives an upper bound on approximation error by choosing a single arbitrary column subset $A_J$. Lemma 2.2 is our main technical lemma.

**Lemma 2.1.** *If $J$ satisfies $\det(V_J) \neq 0$, then the approximation error of $A_J$ can be upper bounded by*

$$\mathsf{Err}^p(A_J) \leq \|\Delta - \Delta_J V_J^{-1} V\|_p^p.$$

**Lemma 2.2.** *Let $S = \{s_{ij}\} \in \mathbb{C}^{k \times m}$ be a complex matrix, $r = (r_1, \cdots, r_m)$ be $m$-dimensional complex vector, and $T = \begin{bmatrix} r \\ S \end{bmatrix} \in \mathbb{C}^{(k+1) \times m}$, then we have*

$$\sum_{I \in [m]^{k+1}} |\det(T_I)|^p \leq C_{p,k} \sum_{l=1}^{m} |r_l|^p \sum_{J \in [m]^k} |\det(S_J)|^p,$$

*where*

$$C_{p,k} = c_{p,k}^p = \begin{cases} (k+1), & 1 \leq p \leq 2, \\ (k+1)^{p-1}, & p \geq 2. \end{cases}$$

We first show that Theorem 1.1 has a clean proof using the two lemmas, as stated below.

*Proof.* of Theorem 1.1: We can WLOG assume that $\mathsf{rank}(V) = k$. In fact, if $\mathsf{rank}(A) < k$, then of course $\mathsf{Err}^p(A_{J^*}) = 0$ and there is nothing to prove. Otherwise if $\mathsf{rank}(A) \geq k$, then by the definition of $V$, we know that $\mathsf{rank}(V) = k$.

We will upper bound the approximation error of the best column by a weighted average of $\mathsf{Err}^p(A_J)$. In other words, we are going to choose a set of non-negative weights $w_J$ such that $\sum_{J\in[m]^k} w_J = 1$, and upper bound $\mathsf{Err}^p(A_{J^*})$ by

$$\mathsf{Err}^p(A_{J^*}) = \min_{J\in[m]^k} \mathsf{Err}^p(A_J) \le \sum_{J\in[m]^k} w_J \mathsf{Err}^p(A_J).$$

In the following analysis, our choice of $w_J$ will be

$$w_J = \frac{|\det(V_J)|^p}{\sum_{I\in[m]^k} |\det(V_I)|^p}.$$

Since $\mathrm{rank}(V) = k$, $w_J$ are well-defined. We first prove

$$|\det(V_J)|^p \mathsf{Err}^p(A_{J^*}) \le \sum_{d=1}^{n}\sum_{i=1}^{m} \left|\det\begin{pmatrix}\Delta_{dL}\\V_L\end{pmatrix}\right|^p, \tag{3}$$

where we denote $L = (i, J) = (i, j_1, \cdots, j_k) \in [m]^{k+1}$.

In fact, when $\det(V_J) = 0$, of course LHS of (3) $= 0 \le$ RHS. When $\det(V_J) \ne 0$, we know that $V_J$ is invertible. By Lemma 2.1,

$$\begin{aligned}
|\det(V_J)|^p \mathsf{Err}^p(A_{J^*}) &\le |\det(V_J)|^p \|\Delta - \Delta_J V_J^{-1} V\|_p^p \\
&= \|\det(V_J)\left(\Delta - \Delta_J V_J^{-1} V\right)\|_p^p \\
&= \sum_{i=1}^{m} \|\det(V_J)\left(\Delta_i - \Delta_J V_J^{-1} V_i\right)\|_p^p \\
&= \sum_{d=1}^{n}\sum_{i=1}^{m} |\det(V_J)\left(\Delta_{di} - \Delta_{dJ} V_J^{-1} V_i\right)|^p \\
&= \sum_{d=1}^{n}\sum_{i=1}^{m} \left|\det\begin{pmatrix}\Delta_{di} & \Delta_{dJ}\\V_i & V_J\end{pmatrix}\right|^p \\
&= \sum_{d=1}^{n}\sum_{i=1}^{m} \left|\det\begin{pmatrix}\Delta_{dL}\\V_L\end{pmatrix}\right|^p.
\end{aligned}$$

The second to last equality follows from the Schur's determinant identity. Therefore (3) holds, and

$$\begin{aligned}
\mathsf{Err}^p(A_{J^*}) &\le \sum_{J\in[m]^k} \frac{|\det(V_J)|^p}{\sum_{I\in[m]^k} |\det(V_I)|^p} \mathsf{Err}^p(A_J) \\
&\le \sum_{d=1}^{n}\left(\frac{1}{\sum_{I\in[m]^k} |\det(V_I)|^p} \sum_{J\in[m]^k}\sum_{i=1}^{m} \left|\det\begin{pmatrix}\Delta_{dL}\\V_L\end{pmatrix}\right|^p\right) \\
&= \sum_{d=1}^{n}\left(\frac{1}{\sum_{I\in[m]^k} |\det(V_I)|^p} \sum_{L\in[m]^{k+1}} \left|\det\begin{pmatrix}\Delta_{dL}\\V_L\end{pmatrix}\right|^p\right).
\end{aligned}$$

By Lemma 2.2,

$$\frac{1}{\sum_{I\in[m]^k} |\det(V_I)|^p} \sum_{L\in[m]^{k+1}} \left|\det\begin{pmatrix}\Delta_{dL}\\V_L\end{pmatrix}\right|^p \le C_{p,k} \sum_{j=1}^{m} |\Delta_{dj}|^p.$$

Therefore,

$$\mathsf{Err}^p(A_{J^*}) \le \sum_{d=1}^{n}\left(C_{p,k} \sum_{j=1}^{m} |\Delta_{dj}|^p\right) = C_{p,k} \sum_{j=1}^{m} \|\Delta_j\|_p^p = C_{p,k}\mathsf{OPT}^p,$$

which means

$$\mathsf{Err}(A_{J^*}) \le C_{p,k}^{1/p}\mathsf{OPT} = c_{p,k}\mathsf{OPT}.$$

$\square$

Therefore, we only need to prove the two lemmas. Lemma 2.1 is relatively easy to prove.

*Proof.* of Lemma 2.1: Recall that by definition of $\Delta_i$, $A_i = UV_i + \Delta_i$,

$$
\begin{aligned}
\mathsf{Err}^p(A_J) &= \min_{Y \in \mathbb{R}^{k \times m}} \|A - A_J Y\|_p^p \\
&\leq \|A - A_J V_J^{-1} V\|_p^p \\
&= \|(UV + \Delta) - (UV_J + \Delta_J)V_J^{-1}V\|_p^p \\
&= \|\Delta - \Delta_J V_J^{-1} V\|_p^p.
\end{aligned}
$$

$\square$

The main difficulty in our analysis comes from Lemma 2.2. The proof is based on Riesz-Thorin interpolation theorem from harmonic analysis. Although the technical details in verifying a key inequality (4) are quite complicated, the remaining part which connects Lemma 2.2 to the Riesz-Thorin interpolation theorem is not that difficult to understand. Below we give a proof to Lemma 2.2 without verifying (4), and leave the complete proof of (4) in the appendix.

*Proof.* of Lemma 2.2: We first state a simplified version of the Riesz-Thorin interpolation theorem, which is the most convenient-to-use version for our proof. The general version can be found in the Appendix.

**Lemma 2.3.** *[Simplified version of Riesz-Thorin] Let $\Lambda : \mathbb{C}^{n_1} \times \mathbb{C}^{n_2} \to \mathbb{C}^{n_0}$ be a multi-linear operator, such that the following inequalities*

$$\|\Lambda(a,b)\|_{p_0} \leq M_{p_0} \|a\|_{p_0} \|b\|_{p_0}.$$

$$\|\Lambda(a,b)\|_{p_1} \leq M_{p_1} \|a\|_{p_1} \|b\|_{p_1}.$$

*hold for all $a \in \mathbb{C}^{n_1}, b \in \mathbb{C}^{n_2}$, then we have*

$$\|\Lambda(a,b)\|_{p_\theta} \leq M_{p_0}^{1-\theta} M_{p_1}^{\theta} \|a\|_{p_\theta} \|b\|_{p_\theta}.$$

*holds for all $\theta \in [0,1]$, where*

$$\frac{1}{p_\theta} := \frac{1-\theta}{p_0} + \frac{\theta}{p_1}.$$

Riesz-Thorin theorem is a classical result in interpolation theory that gives bounds for $L^p$ to $L^q$ operator norm. In general, it is easier to prove estimates within spaces like $L^2$, $L^1$ and $L^\infty$. Interpolation theory enables us to generalize results in those spaces to some $L^p$ and $L^q$ spaces in between with an explicit operator norm. In our application, the $X_i$ is a set of $n_i$ elements and $V_i$ is $\mathbb{C}^{n_i}$, the space of functions on $n_i$ elements.
Now we prove Lemma 2.2. In fact, by symmetricity, Lemma 2.2 is equivalent to

$$
(k+1)! \sum_{I \in \binom{[m]}{k+1}} |\det(T_I)|^p \leq k! C_{p,k} \sum_{t=1}^{m} |r_t|^p \sum_{J \in \binom{[m]}{k}} |\det(S_J)|^p.
$$

Here, $\binom{[m]}{k} = \{(i_1, \cdots, i_k) | 1 \leq i_1 < i_2 < \cdots < i_k \leq m\}$ denotes the $k$-subsets of $[m]$.

Taking $\frac{1}{p}$-th power on both sides, we have the following equivalent form

$$
\left( \sum_{I \in \binom{[m]}{k+1}} |\det(T_I)|^p \right)^{\frac{1}{p}} \leq \frac{c_{p,k}}{(k+1)^{\frac{1}{p}}} \left( \sum_{t=1}^{m} |r_t|^p \right)^{\frac{1}{p}} \left( \sum_{J \in \binom{[m]}{k}} |\det(S_J)|^p \right)^{\frac{1}{p}}.
$$

By Laplace expansion on the first row of $\det(T_I)$, we have for every $I = (i_1, \cdots, i_{k+1}) \in \binom{[m]}{k+1}$

$$
\det(T_I) = \sum_{t=1}^{k+1} (-1)^{t+1} r_{i_t} \det(S_{I_{-t}}).
$$

Here, $I_{-t} = (i_1, \cdots, i_{t-1}, i_{t+1}, \cdots, i_{k+1}) \in \binom{[m]}{k}$.

This motivates us to define the following multilinear map $\Lambda : \mathbb{C}^{\binom{[m]}{1}} \times \mathbb{C}^{\binom{[m]}{k}} \to \mathbb{C}^{\binom{[m]}{k+1}}$: for all $\{a_t\}_{t \in \binom{[m]}{1}} \in \mathbb{C}^{\binom{[m]}{1}}, \{b_J\}_{J \in \binom{[m]}{k}} \in \mathbb{C}^{\binom{[m]}{k}}$, and index set $I = (i_1, \cdots, i_{k+1}) \in \binom{[m]}{k+1}$, $[\Lambda(a, b)]_I$ is defined as

$$[\Lambda(a, b)]_I = \sum_{t=1}^{k+1} (-1)^{t+1} a_{i_t} b_{I_{-t}}.$$

Now, by letting $a_t = r_t, b_J = \det(S_J)$, the inequality can be written as

$$\|\Lambda(a, b)\|_p \le \frac{c_{p,k}}{(k+1)^{\frac{1}{p}}} \|a\|_p \|b\|_p = \max\left(1, (k+1)^{1 - \frac{2}{p}}\right) \|a\|_p \|b\|_p. \tag{4}$$

Let $M_p = \max\left(1, (k+1)^{1 - \frac{2}{p}}\right)$, the inequality can be rewritten as $\|\Lambda(a, b)\|_p \le M_p \|a\|_p \|b\|_p$. We denote

$$\frac{1}{p} = \frac{1 - \theta}{p_0} + \frac{\theta}{p_1}.$$

here, when $p \in [1, 2)$, we choose $p_0 = 1, p_1 = 2$; when $p \in [2, +\infty)$, we choose $p_0 = 2, p_1 = +\infty$. Then, we can observe the following nice property about $M_p$:

$$M_p = M_{p_0}^{1-\theta} M_{p_1}^{\theta} \tag{5}$$

This is exactly the same form as Riesz-Thorin Theorem! Hence, we *only* need to show (4) holds for $p = 1, 2, \infty$, then applying Riesz-Thorin proves all the intermediate cases $p \in (1, 2) \cup (2, \infty)$ immediately.

We leave the complete proof of (4) in the appendix. $\qquad\square$

## 3 Lower Bounds

In this section, we give a proof sketch of Theorem 1.2. The proof is constructive: we prove the theorem by showing for all $\varepsilon > 0$, we can construct a matrix $A(\varepsilon)$, such that selecting every $k$ columns of $A(\varepsilon)$ leads to an approximation ratio at least $\frac{(k+1)^{1 - \frac{1}{p}}}{1 + o_\varepsilon(1)}$. Then, the theorem follows by letting $\varepsilon \to 0^+$. Our choice of $A(\varepsilon)$ is a perturbation of Hadamard matrices.

Throughout the proof, we assume that $k = 2^r - 1$, for some $r \in \mathbb{Z}^+$, and $\varepsilon > 0$ is an arbitrarily small constant. We consider the well known Hadamard matrix of order $(k+1) = 2^r$, defined below:

$$H^{(1)} = 1,$$

$$H^{(2^l)} = \begin{pmatrix} H^{(2^{l-1})} & H^{(2^{l-1})} \\ H^{(2^{l-1})} & -H^{(2^{l-1})} \end{pmatrix}, l \ge 1.$$

Now we can define $A(\varepsilon)$, the construction of lower bound instance: it is a perturbation of $H$ by replacing all the entries on the first row by $\varepsilon$, i.e.,

$$A(\varepsilon)_{ij} = \begin{cases} \varepsilon & \text{when} \quad i = 1, \\ H_{ij} & \text{when} \quad i \ne 1. \end{cases} \tag{6}$$

We can see that $A(\varepsilon)$ is close to a rank-$k$ matrix. In fact, $A(0)$ has rank at most $k$. Therefore, we can upper bound OPT by

$$\text{OPT} \le \|A(\varepsilon) - A(0)\|_p = ((k+1)\varepsilon^p)^{1/p} = (k+1)^{1/p}\varepsilon. \tag{7}$$

The remaining work is to give a lower bound on the approximation error using any $k$ columns. For simplicity of notations, we use $A$ as shorthand for $A(\varepsilon)$ when it's clear from context. Say we

---

**Algorithm 2** [1] SELECTCOLUMNS $(k, A)$: Selecting $O(k \log m)$ columns of $A$.

---

 1: **Input:** Data matrix $A$ and rank parameter $k$.
 2: **Output:** $O(k \log m)$ columns of $A$
 3: **if** number of columns of $A \le 2k$ **then**
 4:     return all the columns of $A$
 5: **else**
 6:     **repeat**
 7:         Let $R$ be uniform at random $2k$ columns of $A$
 8:     **until** at least $(1/10)$-fraction columns of $A$ are $\lambda_p$-approximately covered
 9:     Let $A_{\overline{R}}$ be the columns of $A$ not approximately covered by $R$
10:     return $A_R \cup$ SELECTCOLUMNS $(k, A_{\overline{R}})$
11: **end if**

---

**Algorithm 3** [1]An algorithm that transforms an $O(k \log m)$-rank matrix factorization into a $k$-rank matrix factorization without inflating the error too much.

---

 1: **Input:** $U \in \mathbb{R}^{n \times O(k \log m)}$, $V \in \mathbb{R}^{O(k \log m) \times m}$
 2: **Output:** $W \in \mathbb{R}^{n \times k}$, $Z \in \mathbb{R}^{k \times m}$
 3: Apply Lemma E.1 to $U$ to obtain matrix $W^0$
 4: Run $\ell_p$ linear regression over $Z^0$, s.t. $\|W^0 Z^0 - UV\|_p$ is minimized
 5: Apply Algorithm 1 with input $(Z^0)^T \in \mathbb{R}^{n \times O(k \log m)}$ and $k$ to obtain $X$ and $Y$
 6: Set $Z \leftarrow X^T$
 7: Set $W \leftarrow W^0 Y^T$
 8: Output $W$ and $Z$

---

are using all $(k+1)$ columns except the $j$-th, i.e. the column subset is $A_{[k+1]-\{j\}}$. Obviously, we achieve zero error on all the columns other than the $j$-th. Therefore, the approximation error is essentially the $\ell_p$ distance from $A_j$ to span $\{A_{[k+1]-\{j\}}\}$. We can show that the $\ell_p$ projection from $A_j$ to span $\{A_{[k+1]-\{j\}}\}$ is very close to $\sum_{i \neq j}(-A_i)$, in other words,

$$\mathsf{Err}(A_{[k+1]-\{j\}}) = (1 - o(1))\|A_j - \sum_{i \neq j}(-A_i)\|_p = (1 - o(1))(k+1)\varepsilon \tag{8}$$

The theorem follows by combining (7) and (8). The complete proof can be found in the appendix.

## 4   Analysis of Efficient Algorithms

One drawback of the column subset selection algorithm is its time complexity - it requires $O(m^k \operatorname{poly}(n))$ time, which is not desirable since it's exponential in $k$. However, several more efficient algorithms [1] are designed based on it. Our tighter analysis on Algorithm 1 implies better approximation guarantees on these algorithms as well. The improved bounds can be stated as follows:

**Theorem 4.1.** *Algorithm 2, which runs in* $\operatorname{poly}(m, n, k)$ *time and selects* $k \log m$ *columns, is a bi-criteria* $O(c_{p,k}) = O((k+1)^{\max(1/p, 1-1/p)})$ *approximation algorithm.*

**Theorem 4.2.** *Algorithm 3, which runs in* $\operatorname{poly}(m, n)$ *time as long as* $k = O(\frac{\log n}{\log \log n})$ *is an* $O(c_{p,k}^3 k \log m) = O(k^{\max(1+3/p, 4-3/p)} \log m)$ *approximation algorithm.*

These results improve the previous $O(k)$ and $O(k^4 \log(m))$ bounds respectively. We include the analysis of Algorithm 2 and Algorithm 3 in Appendix for completeness.

#### Acknowledgments

C.D. and P.R. acknowledge the support of Rakuten Inc., and NSF via IIS1909816. The authors would also like to acknowledge two MathOverflow users, known to us only by their usernames, 'fedja' and 'Mahdi', for informing us the Riesz-Thorin interpolation theorem.

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
