[Supplementary Material]

## A  Riesz-Thorin Interpolation Theorem

**Lemma A.1** (Riesz-Thorin interpolation theorem , see Lemma 8.5 in [? ]). *Let $(X_i, \mathfrak{M}_i, \mu_i)$, $i = 0, 1, 2, \cdots, n$ be measure spaces. Let $V_i$ represent the complex vector space of simple functions on $X_i$. Suppose that*

$$\Lambda : V_1 \times V_2 \times \cdots \times V_n \to V_0.$$

*is a multi-linear operator of types $p_0$ and $p_1$ where $p_0, p_1 \in [1, \infty]$, with constants $M_0$ and $M_1$, respectively. i.e.,*

$$\|\Lambda(f_1, f_2, \cdots, f_n)\|_{p_i} \le M_{p_i} \|f_1\|_{p_i} \|f_2\|_{p_i} \cdots \|f_n\|_{p_i}.$$

*for $i = 0, 1$. Let $\theta \in [0, 1]$ and define*

$$\frac{1}{p_\theta} := \frac{1 - \theta}{p_0} + \frac{\theta}{p_1}.$$

*Then, $\Lambda$ is of type $p_\theta$ with constant $M_{p_\theta} := M_{p_0}^{1-\theta} M_{p_1}^{\theta}$, that is,*

$$\|\Lambda(f_1, f_2, \cdots, f_n)\|_{p_\theta} \le M_{p_\theta} \|f_1\|_{p_\theta} \|f_2\|_{p_\theta} \cdots \|f_n\|_{p_\theta}.$$

Lemma **??** is a direct corollary of this theorem.

## B  Lower Bounds

In this section, we will prove Theorem **??**. The proof is constructive: we prove the theorem by showing for all $\varepsilon > 0$, we can construct a matrix $A(\varepsilon)$, such that selecting every $k$ columns of $A(\varepsilon)$ leads to an approximation ratio at least $\frac{(k+1)^{1-\frac{1}{p}}}{(1+k\varepsilon^q)^{1/q}}$. Then, the theorem follows by letting $\varepsilon \to 0^+$. Our choice of $A(\varepsilon)$ is a perturbation of Hadamard matrices, defined below.

Throughout the proof of Theorem **??**, we assume that $k = 2^r - 1$, for some $r \in \mathbb{Z}^+$, and $\varepsilon > 0$ is an arbitrarily small constant.

*Proof.* of Theorem **??**: We consider the well known Hadamard matrix of order $(k + 1) = 2^r$, defined below:

$$H^{(1)} = 1,$$

$$H^{(2^l)} = \begin{pmatrix} H^{(2^{l-1})} & H^{(2^{l-1})} \\ H^{(2^{l-1})} & -H^{(2^{l-1})} \end{pmatrix}, l \ge 1.$$

The Hadamard matrix has the following properties: (we will use $H$ to represent $H^{(2^r)}$ when it's clear from context)

- $H_{di} = 1$ or $H_{di} = -1$.

- All entries on the first row are ones, i.e. $H_{1j} = 1$.

- The columns of $H$ are pairwisely orthogonal, i.e.

$$\sum_{d=1}^{k+1} H_{di} H_{dj} = 0$$

  holds when $i \neq j$.

Now we can define $A(\varepsilon)$: it is a perturbation of $H$ by replacing all the entries on the first row by $\varepsilon$, i.e.,

$$A(\varepsilon)_{ij} = \begin{cases} \varepsilon & \text{when} \quad i = 1, \\ H_{ij} & \text{when} \quad i \neq 1. \end{cases} \tag{1}$$

We can see that $A(\varepsilon)$ is close to a rank-$k$ matrix. In fact, $A(0)$ has rank at most $k$. Also, $A(0)$ is an $2^r \times 2^r$, or equivalently, $(k + 1) \times (k + 1)$ matrix, and it has all zeros on the first row. Therefore, we can upper bound OPT by

$$\mathsf{OPT} \le \|A(\varepsilon) - A(0)\|_p = ((k+1)\varepsilon^p)^{1/p} = (k+1)^{1/p}\varepsilon.$$

The remaining work is to give a lower bound on the approximation error using any $k$ columns. For simplicity of notations, we use $A$ as shorthand for $A(\varepsilon)$ when it's clear from context. Say we are using all $(k+1)$ columns except the $j$-th, i.e. the column subset is $A_{[k+1]-\{j\}}$. Obviously, we achieve zero error on all the columns other than the $j$-th. Therefore, the approximation error is essentially the $\ell_p$ distance from $A_j$ to span $\{A_{[k+1]-\{j\}}\}$. Thus,

$$
\begin{aligned}
\mathsf{Err}(A_{[k+1]-\{j\}}) &= \inf_{x_i \in \mathbb{R}} \left\| A_j - \sum_{i \neq j} x_i A_i \right\|_p \\
&= \inf_{x_i \in \mathbb{R}} \left( \sum_{d=1}^{k+1} \left| A_{dj} - \sum_{i \neq j} x_i A_{di} \right|^p \right)^{1/p} \\
&= \inf_{x_i \in \mathbb{R}} \left( \varepsilon^p \left| 1 - \sum_{i \neq j} x_i \right|^p + \sum_{d=2}^{k+1} \left| H_{dj} - \sum_{i \neq j} x_i H_{di} \right|^p \right)^{1/p}.
\end{aligned}
$$

By Hölder's inequality,

$$
\begin{aligned}
&\left( \varepsilon^p \left| 1 - \sum_{i \neq j} x_i \right|^p + \sum_{d=2}^{k+1} \left| H_{dj} - \sum_{i \neq j} x_i H_{di} \right|^p \right)^{1/p} \left( \left( \frac{1}{\varepsilon} \right)^q + \sum_{d=2}^{k+1} |H_{dj}|^q \right)^{1/q} \\
&\geq \left( 1 - \sum_{i \neq j} x_i \right) + \sum_{d=2}^{k+1} H_{dj} \left( H_{dj} - \sum_{i \neq j} x_i H_{di} \right)
\end{aligned}
$$

where $\frac{1}{p} + \frac{1}{q} = 1$.

We can actually show that $RHS = k + 1$.

Using the fact that $H_{1i} = H_{1j} = 1$ and $\sum_{d=1}^{k+1} H_{di} H_{dj} = 0$,

$$
\begin{aligned}
RHS &= \left( 1 - \sum_{i \neq j} x_i \right) + \sum_{d=2}^{k+1} H_{dj} \left( H_{dj} - \sum_{i \neq j} x_i H_{di} \right) \\
&= \left( 1 - \sum_{i \neq j} H_{1i} H_{1j} x_i \right) + \sum_{d=2}^{k+1} \left( 1 - \sum_{i \neq j} x_i H_{di} H_{dj} \right) \\
&= \sum_{d=1}^{k+1} \left( 1 - \sum_{i \neq j} x_i H_{di} H_{dj} \right) \\
&= (k+1) - \sum_{i \neq j} x_i \left( \sum_{d=1}^{k+1} H_{di} H_{dj} \right) \\
&= k + 1.
\end{aligned}
$$

Now we can finally bound the approximation error

$$\mathsf{Err}(A_{[k+1]-\{j\}}) = \inf_{x_i \in \mathbb{R}} \left( \varepsilon^p \left| 1 - \sum_{i \neq j} x_i \right|^p + \sum_{d=2}^{k+1} \left| H_{dj} - \sum_{i \neq j} x_i H_{di} \right|^p \right)^{1/p}$$

$$\geq \frac{k+1}{\left( \left(\frac{1}{\varepsilon}\right)^q + \sum_{d=2}^{k+1} |H_{dj}|^q \right)^{1/q}}$$

$$= \frac{k+1}{(\varepsilon^{-q} + k)^{1/q}}$$

$$= \frac{(k+1)\varepsilon}{(1 + k\varepsilon^q)^{1/q}}.$$

Thus,

$$\frac{\mathsf{Err}(A_{[k+1]-\{j\}})}{\mathsf{OPT}} \geq \frac{(k+1)^{1-\frac{1}{p}}}{(1 + k\varepsilon^q)^{1/q}}.$$

Note that this bound can be arbitrarily close to $(k+1)^{1-\frac{1}{p}}$ when $\varepsilon$ is small enough, thus we complete the proof. $\qquad\square$

## C  Proof of Equation (??)

Now we are going to prove (??). First, we need to extend the definition of $b_J$ for all $J = (j_1, \cdots, j_k) \in [m]^k$. This definition is similar to the property of determinants.

- When $1 \leq j_1 < j_2 < \cdots < j_k \leq m$, i.e. $J \in \binom{[m]}{k}$, $b_J$ is already defined.
- When there exists $s \neq t, j_s = j_t$, define $b_J = 0$.
- Otherwise, there exists $1 \leq j'_1 < j'_2 < \cdots < j'_k \leq m$ and a permutation $\pi$, such that

$$(j_1, \cdots, j_k) = \pi(j'_1, j'_2, \cdots, j'_k).$$

  Let $J' = (j'_1, j'_2, \cdots, j'_k)$. In such case, we define

$$b_J = \mathrm{sign}(\pi) b_{J'},$$

  where $\mathrm{sign}(\pi)$ is the parity of $\pi$, i.e. $\mathrm{sign}(\pi) = 1$ if $\pi$ is an even permutation, and $\mathrm{sign}(\pi) = -1$ otherwise.

Note that if $J$ is a transposition (2-element exchanges) of $\tilde{J}$, then $b_J = -b_{\tilde{J}}$.

We can also define $[\Lambda(a,b)]_I$ for all $I \in [m]^{k+1}$, by

$$[\Lambda(a,b)]_I = \sum_{t=1}^{k+1} (-1)^{t+1} a_{i_t} b_{I_{-t}}.$$

Here, $I_{-t} = (i_1, \cdots, i_{t-1}, i_{t+1}, \cdots, i_{k+1}) \in [m]^k$. Similarly, if $I$ is a transposition (2-element exchanges) of $\tilde{I}$, then $[\Lambda(a,b)]_I = -[\Lambda(a,b)]_{\tilde{I}}$.

As mentioned before, we only need to verify (??) for the special cases $p = 1, 2, \infty$. In the proof below, we will use either ordered subsets (e.g. $I \in [m]^k$) or unordered subsets (e.g. $I \in \binom{[m]}{k}$), whichever is more convenient.

**Case 1:** $p = 1$. The inequality is equivalent to

$$\|\Lambda(a,b)\|_1 \leq \|a\|_1 \|b\|_1.$$

In fact, by the definition, we always have

$$\|\Lambda(a,b)\|_1 = \sum_{I \in \binom{[m]}{k+1}} |[\Lambda(a,b)]_I| = \frac{1}{(k+1)!} \sum_{I \in [m]^{k+1}} |[\Lambda(a,b)]_I|.$$

Therefore,

$$\|\Lambda(a,b)\|_1 = \frac{1}{(k+1)!} \sum_{I\in[m]^{k+1}} \left|\sum_{t=1}^{k+1}(-1)^{t+1}a_{i_t}b_{I_{-t}}\right|$$

$$\leq \frac{1}{(k+1)!} \sum_{I\in[m]^{k+1}} \sum_{t=1}^{k+1}|a_{i_t}||b_{I_{-t}}|$$

$$= \frac{1}{(k+1)!}(k+1) \sum_{I\in[m]^{k+1}} |a_{i_1}||b_{I_{-1}}|$$

$$= \frac{1}{k!} \sum_{i_1\in[m]} |a_{i_1}| \sum_{J\in[m]^k} |b_J|$$

$$= \sum_{i_1\in[m]} |a_{i_1}| \sum_{J\in\binom{[m]}{k}} |b_J|$$

$$= \|a\|_1\|b\|_1.$$

**Case 2:** $p = \infty$. The inequality is equivalent to

$$\|\Lambda(a,b)\|_\infty \leq (k+1)\|a\|_\infty\|b\|_\infty$$

$$\|\Lambda(a,b)\|_\infty = \max_{I\in\binom{[m]}{k+1}} |[\Lambda(a,b)]_I|$$

$$= \max_{I\in\binom{[m]}{k+1}} \left|\sum_{t=1}^{k+1}(-1)^{t+1}a_{i_t}b_{I_{-t}}\right|$$

$$\leq \max_{I\in\binom{[m]}{k+1}} \sum_{t=1}^{k+1}|a_{i_t}||b_{I_{-t}}|$$

$$\leq \sum_{t=1}^{k+1} \max_{i_t\in[m]} |a_{i_t}| \max_{J\in\binom{[m]}{k}} |b_J|$$

$$= (k+1) \max_{i_1\in[m]} |a_{i_1}| \max_{J\in\binom{[m]}{k}} |b_J|$$

$$= (k+1)\|a\|_\infty\|b\|_\infty.$$

**Case 3:** $p = 2$. The inequality is equivalent to

$$\|\Lambda(a,b)\|_2 \leq \|a\|_2\|b\|_2.$$

$$\|\Lambda(a,b)\|_2^2 = \sum_{I\in\binom{[m]}{k+1}} |[\Lambda(a,b)]_I|^2$$

$$= \frac{1}{(k+1)!} \sum_{I\in[m]^{k+1}} |[\Lambda(a,b)]_I|^2$$

$$= \frac{1}{(k+1)!} \sum_{I\in[m]^{k+1}} \left|\sum_{t=1}^{k+1}(-1)^{t+1}a_{i_t}b_{I_{-t}}\right|^2.$$

Note that

$$\left|\sum_{t=1}^{k+1}(-1)^{t+1}a_{i_t}b_{I_{-t}}\right|^2 = \left(\sum_{t=1}^{k+1}(-1)^{t+1}a_{i_t}b_{I_{-t}}\right)\left(\sum_{s=1}^{k+1}(-1)^{s+1}\bar{a}_{i_s}\bar{b}_{I_{-s}}\right)$$

$$= \sum_{t=1}^{k+1}|a_{i_t}|^2|b_{I_{-t}}|^2 + \sum_{1\leq t\neq s\leq k+1}(-1)^{t+s}a_{i_t}b_{I_{-t}}\bar{a}_{i_s}\bar{b}_{I_{-s}}.$$

Therefore,

$$(k+1)!\|\Lambda(a,b)\|_2^2 = \sum_{I\in[m]^{k+1}}\sum_{t=1}^{k+1}|a_{i_t}|^2|b_{I_{-t}}|^2 + \sum_{I\in[m]^{k+1}}\sum_{1\leq t\neq s\leq k+1}(-1)^{t+s}a_{i_t}b_{I_{-t}}\bar{a}_{i_s}\bar{b}_{I_{-s}}.$$

The first term can be simplified as

$$\sum_{I\in[m]^{k+1}}\sum_{t=1}^{k+1}|a_{i_t}|^2|b_{I_{-t}}|^2$$

$$= (k+1)\sum_{i_1\in[m]}|a_{i_1}|^2\sum_{J\in[m]^k}|b_J|^2$$

$$= (k+1)!\|a\|_2^2\|b\|_2^2.$$

Therefore, we only need to prove that the second term is non-positive.

When $t<s$,

$$b_{I_{-s}} = b_{(i_1,\cdots,i_{s-1},i_{s+1},\cdots,i_{k+1})}$$
$$= (-1)^{t-1}b_{(i_l,i_1,\cdots,i_{t-1},i_{t+1},\cdots,i_{s-1},i_{s+1},\cdots,i_{k+1})}$$
$$= (-1)^{t-1}b_{(i_t,I_{-\{t,s\}})},$$

and

$$b_{I_{-t}} = b_{(i_1,\cdots,i_{t-1},i_{t+1},\cdots,i_{k+1})}$$
$$= (-1)^{s-2}b_{(i_s,i_1,\cdots,i_{t-1},i_{t+1},\cdots,i_{s-1},i_{s+1},\cdots,i_{k+1})}$$
$$= (-1)^{s-2}b_{(i_s,I_{-\{t,s\}})}.$$

Therefore,

$$(-1)^{t+s}b_{I_{-t}}\bar{b}_{I_{-s}} = -b_{(i_s,I_{-\{t,s\}})}\bar{b}_{(i_t,I_{-\{t,s\}})}.$$

The same argument holds for the case $t>s$. Thus, for each pair of $(t,s)$, we have

$$\sum_{I\in[m]^{k+1}}(-1)^{t+s}a_{i_t}b_{I_{-t}}\bar{a}_{i_s}\bar{b}_{I_{-s}}$$

$$= -\sum_{I\in[m]^{k+1}}a_{i_t}\bar{a}_{i_s}b_{(i_s,I_{-\{t,s\}})}\bar{b}_{(i_t,I_{-\{t,s\}})}$$

$$= -\sum_{J\in[m]^{k-1}}\sum_{i_t=1}^m\sum_{i_s=1}^m a_{i_t}\bar{a}_{i_s}b_{(i_s,J)}\bar{b}_{(i_t,J)}$$

$$= -\sum_{J\in[m]^{k-1}}\left(\sum_{i_t=1}^m a_{i_t}\bar{b}_{(i_t,J)}\right)\left(\sum_{i_s=1}^m \bar{a}_{i_s}b_{(i_s,J)}\right)$$

$$= -\sum_{J\in[m]^{k-1}}\left|\sum_{i_t=1}^m a_{i_t}\bar{b}_{(i_t,J)}\right|^2.$$

Thus, the second term can be simplified as

$$\sum_{I \in [m]^{k+1}} \sum_{1 \leq t \neq s \leq k+1} (-1)^{t+s} a_{i_t} b_{I_{-t}} \bar{a}_{i_s} \bar{b}_{I_{-s}}$$

$$= \sum_{1 \leq t \neq s \leq k+1} \sum_{I \in [m]^{k+1}} (-1)^{t+s} a_{i_t} b_{I_{-t}} \bar{a}_{i_s} \bar{b}_{I_{-s}}$$

$$= -2k(k+1) \sum_{J \in [m]^{k-1}} \left| \sum_{i_t=1}^{m} a_{i_t} \bar{b}_{(i_l, J)} \right|^2 \leq 0.$$

$\square$

## D    Analysis for A $\mathsf{poly}(nm)$-Time Bi-Criteria Algorithm

We can prove that Algorithm **??** from [**?** ] runs in time $\mathsf{poly}(nm)$ but returns $O(k \log m)$ columns of $A$ that can be used in place of $U$, with an error $O(c_{p,k})$ times the error of the best $k$-factorization. In other words, it obtains more than $k$ columns but achieves a polynomial running time. The analysis can derived by slightly modifying the definition and proof in [**?** ].

**Definition D.1** (Approximate coverage). *Let $S$ be a subset of $k$ column indices. We say that column $A_i$ is $\lambda_p$-**approximately covered** by $S$ if for $p \in [1, \infty)$ we have $\min_{x \in \mathbb{R}^{k \times 1}} \|A_S x - A_i\|_p^p \leq \lambda \frac{100 c_{p,k}^p \|\Delta\|_p^p}{n}$, and for $p = \infty$, $\min_{x \in \mathbb{R}^{k \times 1}} \|A_S x - A_i\|_\infty \leq \lambda(k+1)\|\Delta\|_\infty$. If $\lambda = 1$, we say $A_i$ is* **covered** *by $S$.*

We first show that if we select a set $R$ columns of size $2k$ uniformly at random in $\binom{[m]}{2k}$, with constant probability we cover a constant fraction of columns of $A$.

**Lemma D.1.** *Suppose $R$ is a set of $2k$ uniformly random chosen columns of $A$. With probability at least $2/9$, $R$ covers at least a $1/10$ fraction of columns of $A$.*

*Proof.* Same as the proof of Lemma 6 in [**?** ] except that we use $c_{p,k}^p$ instead of $(k+1)$ in the approximation bounds. $\square$

We are now ready to introduce Algorithm **??**. As mentioned in [**?** ], we can without loss of generality assume that the algorithm knows a number $N$ for which $|\Delta|_p \leq N \leq 2|\Delta|_p$. Indeed, such a value can be obtained by first computing $|\Delta|_2$ using the SVD. Note that although one does not know $\Delta$, one does know $|\Delta|_2$ since this is the Euclidean norm of all but the top $k$ singular values of $A$, which one can compute from the SVD of $A$. Then, note that for $p < 2$, $|\Delta|_2 \leq |\Delta|_p \leq n^{2-p}|\Delta|_2$, while for $p \geq 2$, $|\Delta|_p \leq |\Delta|_2 \leq n^{1-2/p}|\Delta|_p$. Hence, there are only $O(\log n)$ values of $N$ to try, given $|\Delta|_2$, one of which will satisfy $|\Delta|_p \leq N \leq 2|\Delta|_p$. One can take the best solution found by Algorithm **??** for each of the $O(\log n)$ guesses to $N$.

**Theorem D.1.** *With probability at least $9/10$, Algorithm **??** runs in time $\mathsf{poly}(nm)$ and returns $O(k \log m)$ columns that can be used as a factor of the whole matrix inducing $\ell_p$ error $O(c_{p,k}|\Delta|_p)$.*

*Proof.* Same as the proof of Theorem 7 in [**?** ] except that we use $c_{p,k}^p$ instead of $(k+1)$ in the approximation bounds. $\square$

## E    Analysis for A $((k \log n)^k \mathsf{poly}(mn))$-Time Algorithm

In this section we show how to get a rank-$k$, $O(c_{p,k}^3 k \log m)$-approximation efficiently starting from a rank-$O(k \log m)$ approximation. This algorithm runs in polynomial time as long as $k = O\left(\frac{\log n}{\log \log n}\right)$.

Let $U$ be the columns of $A$ selected by Algorithm **??**.

### E.1 An Isoperimetric Transformation

The first step of the proof is to show that we can modify the selected columns of $A$ to span the same space but to have small distortion. For this, we need the following notion of isoperimetry.

**Definition E.1** (Almost isoperimetry). *A matrix $B \in \mathbb{R}^{n \times m}$ is* almost-$\ell_p$-isoperimetric *if for all $x$, we have*

$$\frac{\|x\|_p}{2m} \leq \|Bx\|_p \leq \|x\|_p.$$

The following lemma from [**?**] show that given a full rank $A \in \mathbb{R}^{n \times m}$, it is possible to construct in polynomial time a matrix $B \in \mathbb{R}^{n \times m}$ such that $A$ and $B$ span the same space and $B$ is almost-$\ell_p$-isoperimetric.

**Lemma E.1** (Lemma 10 in [**?**]). *Given a full (column) rank $A \in \mathbb{R}^{n \times m}$, there is an algorithm that transforms $A$ into a matrix $B$ such that* $\mathrm{span}\,\{A\} = \mathrm{span}\,\{B\}$ *and $B$ is almost-$\ell_p$-isoperimetric. Furthermore the running time of the algorithm is $poly(nm)$.*

### E.2 Reducing the Rank to $k$

Here we give an analysis of Algorithm **??** from [**?**]. It reduces the rank of our low-rank approximation from $O(k \log m)$ to $k$. Let $\delta = \|\Delta\|_p = \mathsf{OPT}$.

**Theorem E.1.** *Let $A \in \mathbb{R}^{n \times m}$, $U \in \mathbb{R}^{n \times O(k \log m)}$, $V \in \mathbb{R}^{O(k \log m) \times m}$ be such that $\|A - UV\|_p = O(k\delta)$. Then, Algorithm **??** runs in time $O(k \log m)^k (mn)^{O(1)}$ and outputs $W \in \mathbb{R}^{n \times k}$, $Z \in \mathbb{R}^{k \times m}$ such that $\|A - WZ\|_p = O((c_{p,k}^3 k \log m)\delta)$.*

*Proof.* We start by bounding the running time. Step 3 is computationally the most expensive since it requires to execute a brute-force search on the $O(k \log m)$ columns of $(Z^0)^T$. So the running time is $O((k \log m)^k (mn)^{O(1)})$ .

Now we have to show that the algorithm returns a good approximation. The main idea behind the proof is that $UV$ is a low-rank approximable matrix. So after applying Lemma E.1 to $U$ to obtain a low-rank approximation for $UV$ we can simply focus on $Z^0 \in \mathbb{R}^{O(k \log m) \times n}$. Next, by applying Algorithm **??** to $Z^0$, we obtain a low-rank approximation in time $O(k \log m)^k (mn)^{O(1)}$. Finally we can use this solution to construct the solution to our initial problem.

We know by assumption that $\|A - UV\|_p = O(c_{p,k}\delta)$. Therefore, it suffices by the triangle inequality to show $\|UV - WZ\|_p = O(c_{p,k}^3 k \log m\delta)$. First note that $UV = W^0 Z^0$ since Lemma E.1 guarantees that $\mathrm{span}\,\{U\} = \mathrm{span}\,\{W^0\}$. Hence we can focus on proving $\|W^0 Z^0 - WZ\|_p \leq O((c_{p,k}^3 k \log m)\delta)$.

We first prove two useful intermediate steps.

**Lemma E.2.** *There exist matrices $U^* \in \mathbb{R}^{n \times k}$, $V^* \in \mathbb{R}^{k \times m}$ such that $\|W^0 Z^0 - U^* V^*\|_p = O(c_{p,k}\delta)$.*

*Proof.* Same as the proof of Lemma 12 in [**?**] except that we use $O(c_{p,k}\delta)$ instead of $O(k\delta)$. $\square$

**Lemma E.3.** *There exist matrices $F \in \mathbb{R}^{O(k \log m) \times k}$, $D \in \mathbb{R}^{k \times n}$ such that $\|W^0(Z^0 - FD)\|_p = O(c_{p,k}^2 \delta)$.*

*Proof.* Same as the proof of Lemma 13 in [**?**] except that we use $O(c_{p,k}\delta)$ and $O(c_{p,k}^2 \delta)$instead of $O(k\delta)$ and $O(k^2\delta)$. $\square$

Now from the guarantees of Lemma E.1 we know that for any vector $y$, $\|W^0 y\|_p \leq \frac{\|y\|_p}{k \log m}$. So we have $\|Z^0 - FD\|_p \leq O((c_{p,k}^2 k \log m)\delta)$, Thus $\|(Z^0)^T - D^T F^T\|_p \leq O((c_{p,k}^2 k \log m)\delta)$, so $(Z^0)^T$ has a low-rank approximation with error at most $O((c_{p,k}^2 k \log m)\delta)$. So we can apply Theorem **??** again and we know that there are $k$ columns of $(Z^0)^T$ such that the low-rank approximation obtained starting from those columns has error at most $O((c_{p,k}^3 k \log m)\delta)$. We obtain such a low-rank

approximation from Algorithm **??** with input $(Z^0)^T \in \mathbb{R}^{n \times O(k \log m)}$ and $k$. More precisely, we obtain an $X \in \mathbb{R}^{n \times k}$ and $Y \in \mathbb{R}^{k \times O(k \log m)}$ such that $\|(Z^0)^T - XY\|_p \leq O((c_{p,k}^3 k \log m)\delta)$. Thus $\|Z^0 - Y^T X^T\|_p \leq O((c_{p,k}^3 k \log m)\delta)$.

Now using again the guarantees of Lemma E.1 for $W^0$, we get $\|W^0(Z^0 - Y^T X^T)\|_p \leq O((c_{p,k}^3 k \log m)\delta)$. So $\|W^0(Z^0 - Y^T X^T)\|_p = \|W^0 Z^0 - WZ)\|_p = \|UV - WZ\|_p \leq O((c_{p,k}^3 k \log m)\delta)$. By combining it with $\|A - UV\|_p = O(c_{p,k}\delta)$ and using the Minkowski inequality, the proof is complete. $\qquad\square$