[Reviews · NeurIPS 2019]

Reviewer 1



The main result is an extension of the two known results: a recent result from [13] that provide the suboptimal approximation ratio of order $O(k+1)$ over $p \ge 1$, and the classical result from Deshpande et al. ([14]) that analyzes the case $p = 2$ (i.e., Frobenius norm), and proves the bound $O(\sqrt{k+1})$ matching that of the present work. The key technical novelty is the application of the Riesz-Torin theorem that allows to bound the approximation ratio for any~$p \ge 1$ by interpolating the bounds for $p = 1$, $p =2$, and the bound for $p = \infty$ obtained by the authors (the analysis for the latter case is, in fact, way simpler than for the case $p = 2$ addressed in [14]). This allows to obtain sharp results for $p \in [2, \infty$, whereas those for $p \in [1,2]$ remain suboptimal. As far as I can tell, the Riesz-Torin theorem has been hitherto unknown in the TCS community, and drawing attention to it would be useful; hence I recommend to accept the paper. The main reason for my (modest) evaluation score is that the case $p = 2$, clearly the hardest to analyze among $p \in \{1,2,\infty\}$, is already known from the classical work [14]. The key contribution is really the application of the Riesz-Torin theorem to interpolate between the different values of $p$; arguably, this step is less interesting mathematically than the analysis of the case $p = 2$ (cf. lines 412-412 vs. lines 421-423 in Appendix 3). The lower bound construction also crucially relies on the known results. However, the application of the Riesz-Torin theorem is non-trivial (as it invoves introducing a specific multi-linear map), so I think this is a decent contribution, and the paper is worth to be accepted, even in absence of experiments and demonstrated applications of the new results. My other remarks are minor and mostly refer to the exposition. First, the authors denote the usual and ordered subsets by the same symbols $I$ and $J$ throughout the text, which is at times quite confuzing, and hindered my verification of the proof of Theorem 1.1 (which I was still able to verify). For example, in lines 123-127, $I$ and $J$ denote the ordered subsets, whil, e.g., in the proof of Lemma 2.3 the same symbols are used for unordered subsets. Second, the reduction to the cases $p \in \{1,2,\infty\}$ in lines 223-227 is very concise (I had to perform the calculation ad-hoc), as this part of the text was unreadable for me. I urge the authors to improve this part. Regarding the quality of the text and the exposition, the paper is quite well-written. A few typos: L194: "The second [to] last" L417: The factor $2$ in the second term of the right hand side is extraneous, accordingly further down the text. This does not change the argument. L421: $l$ instead of $t$ in the second line of the display.

Reviewer 2



I read the author feedback. Thanks for clarifying how the analysis of the case p=2 is different than prior work. ---------- The paper considers low-rank matrix approximation in l_p norm for any p>=1. One of the popular approaches in this area is to construct a rank-k approximation of an n x m matrix A from a subset of k columns of A. This restriction is natural in practical settings when columns represent data points and we want to find a small but representative core-set of data. While this problem is well understood for p=2, this is not the case for other l_p norms which offer advantages in terms of robustness and sparsity of the solutions. Prior to this work, Chierichetti et al (2017) [13] showed that for any p>=1 a subset of k columns can always be found such that the l_p approximation error is within a k+1 factor of the optimum, while for p=2 Deshpande et al (2006) [14] showed that for p=2 a factor of sqrt{k+1} can be achieved. The results of this paper close that gap by showing that we can achieve a (k+1)^{1/p} factor for 1<=p<=2 and (k+1)^{1-1/p} for p>=2. This matches the best known factors for p=1 and 2 but improves them in all other cases. New matching lower bounds are shown for p>=2. While algorithmic efficiency is of secondary concern in this paper, the main results imply improved guarantees for existing polynomial time algorithms for this task. It is worth noting that the cases of p=1 and 2 are still the ones with primary practical interest, and it would be helpful if the authors found a reference to an applied paper showing the advantages of low-rank approximation with l_p error for, say, 1

Reviewer 3



The paper is well written, well motivated and most of the results are stated clearly with well written proofs. I have enumerated my questions and comments below. Questions and Comments 1. In Ln. 46 the authors say "Unfortunately the loss surface of the problem suffers from spurious local minima and saddle points [12]". [12] doesnt seem to be a correct reference for this fact. Is this fact known? And if yes, can the authors point to a reference for this. 2. (Ln. 70) CUR is undefined. 3. (Ln. 79) I think you mean lower bounded by \Omega(k^{1-2/p}) 4. (Ln. 83) you say there is O(k^{2/p}-1) gap but given your results there seems to be a O(k^{1/p - 1}) gap. 5. Theorem 1.2 is hard for me to understand. Specifically I am not certain I understand what "There exist infinitely many different valyes of k" means here. Do the authors mean for every m there exists a k such that the lower bound holds? 6. (Ln. 126) The dimension of matrix seems non standard, I would prefer if the authors change that to row of the matrix (which is the term that they have used before) ============Post Author Feedback============================= I would like to thank the authors for their polite rebuttal and for clarifying my confusion about their lower bound.

[Author Response · NeurIPS 2019]

We would like to thank the reviewers for the careful and thorough reading of our submission. We are appreciative of the
many suggestions for improvements and insightful questions. In the limited space below, we respond to some of the
main concerns raised by the three reviewers.

**The novelty of our analysis in the case** $p = 2$ **[Reviewer 1 and Reviewer 2]** As the reviewers note, for the case
$p = 2$, the tight approximation bound of $CSS$ is already known in the classic work [14]. However, our work has a
different goal: to provide a unified way of analyzing approximation bounds for all different values of $p$. With this goal
in mind, we introduced the Riesz-Thorin theorem as a general framework, and found that the existing analysis in [14]
unfortunately does not fit in this framework, due to the technical differences stated below.

The main technical difference can be found in line 219-222: the Riesz-Thorin theorem requires that Equation (4)
should hold for *all* $\{a_t\}_{t \in \binom{[m]}{1}} \in \mathbb{C}^{\binom{[m]}{1}}, \{b_J\}_{J \in \binom{[m]}{k}} \in \mathbb{C}^{\binom{[m]}{k}}$. In comparison, to prove that $CSS$ is a $\sqrt{k+1}$
approximation, one only needs to show that the equation holds for $\{a_t\}_{t \in \binom{[m]}{1}} \in \mathbb{C}^{\binom{[m]}{1}}, \{b_J\}_{J \in \binom{[m]}{k}} \in K$, where $K$
is a subset of $\mathbb{C}^{\binom{[m]}{k}}$ defined as $K = \{\{b_J\}_{J \in \binom{[m]}{k}} | b_J = \det(S_J), S \in \mathbb{C}^{k \times m}\}$. It is easy to see that the requirement
of Riesz-Thorin is significantly stronger, since the set $K$ is determined by only $km$ parameters (the matrix $S$), while
$\mathbb{C}^{\binom{[m]}{k}}$ is a $\binom{m}{k}$-dimensional space. In other words, we need a much stronger inequality in order to apply Riesz-Thorin
theorem, hence we provided a brand new proof for the $p = 2$ setting (line 421-423, as mentioned by reviewer 1) which
is completely different from [14].

**The choice of** $p$ **in** $l_p$ **low rank approximation [Reviewer 1 and Reviewer 2]** While our paper is mostly theoretical,
we do believe that choosing an appropriate value of $p$ can make a difference in practice. The $\ell_p$ low rank approximation
problem has attracted interest relatively recently, see for instance ICML 2017 paper [13]. A related problem $\ell_p$ linear
regression however has been studied extensively in the statistics community, and these two problems share similar
motivation. In particular, if we assume a statistical model $A_{ij} = A_{ij}^\star + \varepsilon_{ij}$, where $A^\star$ is a low rank matrix and $\varepsilon_{ij}$ are
i.i.d. noise, the different values of $p$ correspond to the MLE of different noise distributions, say $p = 1$ for Laplacian
noise and $p = 2$ for Gaussian noise. To capture a broader range of realistic noises in complex datasets, it is beneficial to
expand our choices beyond the standard ones $(1, 2, \infty)$.

**Sampling Based Algorithms [Reviewer 2]** This is definitely an important direction for the future works that we
intend to explore. Our determinantal weights are indeed inspired by [14] and we will include the comparison in our final
version. The sampling methods for the $p = 2$ case, e.g. volume sampling, are closely related to the determinantal point
process. To generalize this approach to $\ell_p$ setting, we need an efficient way to implement the exponentiated variant of
volume sampling (i.e. sample a subset $S$ with probability proportional to $\det(V_S)^\alpha$). To the best of our knowledge, this
problem has not been resolved yet. In fact, even computing the normalizing constant of the distribution is difficult - it
was stated as an open problem in Section 7.2 of the survey "Determinantal point processes for machine learning". We
refer the reviewer to the NeurIPS 2018 paper "Exponentiated Strongly Rayleigh Distributions" for recent progress on
this problem.

**Clarification on Theorem 1.2 [Reviewer 3]** We thank the reviewer for pointing out the ambiguity of the informal
statement of the theorem. We proved that there are infinitely many different values of $k$, such that for each $k$, there
exists a matrix $A$ such that CSS cannot do better than $(k+1)^{1-\frac{1}{p}}$ approximation. The dimensions of the matrix $A$ ($m$
and $n$) are not fixed for these different $k$'s.

**Optimization landscape of low rank approximation [Reviewer 3]** We refer the reviewer to the paper "Neural
networks and principal component analysis: Learning from examples without local minima" where the paper shows
that for low-rank approximation problem even as easy as PCA ($p = 2$), saddle points exist. We are happy to modify the
relevant sentence in our paper as "Unfortunately the loss surface of the problem suffers from many saddle points" and
replace [12] with the reference above.

**The optimality gap of the bound when** $1 < p < 2$ **[Reviewer 3]** (line 83) We note that the lower bound (Theorem
1.2) also applies to the case $1 < p < 2$. Therefore, we have an upper bound of $(k+1)^{\frac{1}{p}}$ and a lower bound $(k+1)^{1-\frac{1}{p}}$,
hence we were correct on the $(k+1)^{\frac{2}{p}-1}$ optimality gap.

**Improvements on Exposition [Reviewer 1,2,3]** We thank all three reviewers for their suggestions on the exposition
and will take them into account in the final version. In particular, we will clarify the difference of ordered and unordered
sets; add more explanation to the reduction in the proof of Lemma 2.2; improve the sentence structure in Lemma 2.3
and include the definition of CUR factorization in the introduction.

[Meta-Review · NeurIPS 2019]

This is a solid paper, bringing new results on low rank matrix approximation. It makes an original use of the Riesz-Thorin theorem and provides new contributions on matrix low-rank approximation with L_p norms.